# Involvement of Peripheral Monocytes with IL-1β in the Pathogenesis of West Syndrome

**DOI:** 10.3390/jcm11020447

**Published:** 2022-01-16

**Authors:** Tomoko Takamatsu, Gaku Yamanaka, Koko Ohno, Kanako Hayashi, Yusuke Watanabe, Mika Takeshita, Shinji Suzuki, Shinichiro Morichi, Soken Go, Yu Ishida, Shingo Oana, Yasuyo Kashiwagi, Hisashi Kawashima

**Affiliations:** Department of Pediatrics and Adolescent Medicine, Tokyo Medical University, 6-7-1 Nishi-Shinjuku, Shinjuku, Tokyo 160-0023, Japan; gaku@tokyo-med.ac.jp (G.Y.); koko_deux61116@yahoo.co.jp (K.O.); kanako.hayashi.0110@gmail.com (K.H.); vandersar_0301@yahoo.co.jp (Y.W.); jerryfish_mika@yahoo.co.jp (M.T.); shin.szk@gmail.com (S.S.); s.morichi@gmail.com (S.M.); soupei59@gmail.com (S.G.); ishiyu@tokyo-med.ac.jp (Y.I.); oanas@tokyo-med.ac.jp (S.O.); hoyohoyo@tokyo-med.ac.jp (Y.K.); hisashi@tokyo-med.ac.jp (H.K.)

**Keywords:** West syndrome, infantile spasms, cytokine, developmental, epileptic encephalopathy

## Abstract

Neuroinflammation has been implicated in the pathogenesis of West syndrome (WS). Inflammatory cytokines, including interleukin-1β(IL-1β), have been reported to be associated with epilepsy. However, the assessment of cytokine changes in humans is not always simple or deterministic. This study aimed to elucidate the immunological mechanism of WS. We examined the intracellular cytokine profiles of peripheral blood cells collected from 13 patients with WS, using flow cytometry, and measured their serum cytokine levels. These were compared with those of 10 age-matched controls. We found that the WS group had significantly higher percentages of inter IL-1β, interleukin-1 receptor antagonist (IL-1RA)-positive monocytes, and interferon gamma (IFN-γ) in their CD8+ T cells than the control group. Interestingly, the group with sequelae revealed significantly lower levels of intracellular IFN-γ and IL-6 in their CD8+ T and CD4+ T cells, respectively, than the group without sequelae. There was no correlation between the ratios of positive cells and the serum levels of a particular cytokine in the WS patients. These cytokines in the peripheral immune cells might be involved in the neuroinflammation of WS, even in the absence of infectious or immune disease. Overall, an immunological approach using flow cytometry analysis might be useful for immunological studies of epilepsy.

## 1. Introduction

West syndrome (WS) is a severe epileptic encephalopathy, which causes spasm attacks and hypsarrhythmia on an electroencephalogram in infants and developmental delay in approximately 80–90% of cases [1]. Although its pathogenesis remains obscure, anti-inflammatory therapies, such as adrenal cortisol hormone therapy (ACTH therapy) and gamma globulin, are helpful [2,3,4,5]. In addition, viral infection can improve the clinical symptoms of WS transiently [6,7]. Thus, the existence of neuroinflammation in the pathogenesis of WS has been suggested. In humans, cytotoxic edema derived from inflammatory cytokines has been detected in WS using tensor imaging [8], and pharmacological targeting of the IL-1 receptor pathway has shown a close link between inflammatory cytokines and epileptogenesis in a mouse model [9].

Various cytokines have been recognized as being potentially related to the pathogenesis of epilepsy. In particular, interleukin (IL)-1β is particularly relevant in epileptogenesis [9,10,11,12]. In an experimental mouse model of status epilepticus, IL-1β was found to be elevated after seizures at the mRNA level in hippocampal astrocytes, peaking at 24 h and returning to baseline after 5 days [11]. IL-1β is produced by microglia in the brain and has been reported to influence astrocytes to acquire epileptogenesis [12]. However, these results have not been demonstrated in West’s mouse model, nor has it been studied in humans. Although Shiihara et al. reported decreased levels of IL-1β after ACTH therapy [13], recent studies, including our own, have not yielded equivalent findings and detected no significant differences in IL-1RA levels in a large number of samples [14]. We previously reported that serum IL-1RA levels increased after treatment, following the resolution of clinical symptoms. Some reports suggest that treatment with ACTH increases the number of CD4+ cells and CD3+ cells [15].

There have been several previous reports on cytokines and West syndrome. Tekgul et al. [4] reported that IL-6 was higher in the WS group but was significantly lower than that in the control group, which included patients with meningitis and convulsions. However, a study by Haginoya et al. [5], and a study by our group [14], did not detect significant levels of pro-inflammatory cytokines, including IL-6, TNF-α, and IL-1b, in patients with WS compared with controls. As for the serum, Liu, Wang et al. [3] showed that there were elevated levels of IL-2, TNF-α, and IFN-α, while Ture et al. [16] reported that IL-6 and IL-17A levels were significantly higher in the untreated patient group compared with the healthy control group. To date, there are no conclusive findings on the inflammatory cytokine assessment of WS [2]. Cytokine measurements in humans are not always simple or deterministic; a systematic review also highlighted controversial results [17]. Many reports suggest that seizure timing, sample processing, and blood collection can affect the results of cytokine analysis. The timing of seizures, processing of specimens, and blood drawing may influence cytokine analysis results [18,19,20,21,22].

IL-1β circulating in the blood is particularly unstable. Despite having active systemic auto-inflammatory disease, IL-1β levels may appear within normal limits [23]. In addition, the involvement of IL-1β in WS pathogenesis has not yet been investigated. However, flow cytometric analysis revealed that intracellular IL-1β levels in peripheral monocytes of pediatric patients with intractable epilepsy were higher than those in the controls and were associated with clinical symptoms [10].

Therefore, in this study, we aimed to evaluate the intracellular cytokine profiles and cell surface markers of the cells involved in WS using flow cytometric analysis in order to elucidate the immunological mechanisms of WS. Moreover, we attempted to investigate the association between these immunological assessments and the sequelae of the patients.

## 2. Materials and Methods

### 2.1. Patients

In total, 13 pediatric patients with WS and 10 healthy controls were enrolled retrospectively. We selected inpatients aged 3–7 months that were treated at the Department of Pediatrics and Adolescents, Tokyo Medical University Hospital (Tokyo, Japan), between January 2017 and February 2021. Table 1 summarizes the clinical characteristics of all patients. We defined mild developmental delay as a developmental quotient (DQ) > 70. We also defined good prognosis as normal development and mild developmental delay. DQ was estimated using the Enjoji Scale of Infant Analytical Development, released in 1977. It is widely used in Japan [24] and consists of the following six domains: gross motor skills, fine motor skills, sociality, activities of daily living, language understanding, and speaking ability.

#### 2.1.1. Inclusion Criteria

WS was defined as epileptic spasms in clusters, and the ictal period of hypsarrhythmia was recorded on video electroencephalography [25]. Blood samples were collected during the ictal period before treatment with ACTH or vigabatrin. All patients provided blood samples within 1 month of experiencing epileptic spasms.

The control group comprised age- and sex-matched individuals without inflammatory diseases, including malnourishment and asthma without fever. The children’s parents provided informed consent on behalf of their children for the use of their clinical samples. The study protocol was approved by the Ethics Committee of Tokyo Medical University (approval number: SH3779) and was conducted in accordance with the tenets of the Declaration of Helsinki and its later amendments.

#### 2.1.2. Exclusion Criteria

Patients with inflammatory condition, such as pyrexia, and viral infection were excluded from this study.

### 2.2. Blood Sampling, Isolation, and Activation of Lymphocytes

We collected heparinized venous blood (5 mL) from all participants. Peripheral blood mononuclear cells (PBMC) (5 × 10^6^ cells/mL) were obtained using Ficoll-Paque PLUS (Pharmacia, Uppsala, Sweden) and resuspended in RPMI-1640 supplemented with 10% fetal calf serum. We stained the intracellular effector molecules with brefeldin A (BioLegend, San Diego, CA, USA) and stimulated the lymphocyte samples with phytohemagglutinin (20 ng/mL) and ionomycin (1 µg/mL; Sigma-Aldrich, St. Louis, MO, USA) for 4 h. We also stimulated monocytes with LPS (20 ng/mL; O55:B5 *E. coli*, Merck KGaA, Darmstadt, Germany).

### 2.3. Flow Cytometry Analysis

Based on previous reports [13,26,27], flow cytometric analysis was performed using eight colors, including cell-surface makers CD3, CD4, CD8, CD14, CD25, and CD69 (eBioscience, San Diego, CA, USA and BioLegend, San Diego, CA, USA). Appendix A lists the results of the intracellular cytokine and cell surface marker analysis. For intracellular staining of these cytokines, the treated PBMCs were permeabilized with cell surface antigens, stained, and fixed with the Cytofix/Cytoperm Fixation/Permeabilization Solution (BD Biosciences, San Jose, CA, USA). The samples were subsequently analyzed using the FACS Canto II flow cytometer and the BD FACSDiva Flowjo software (BD Biosciences, Franklin Lakes, NJ, USA).

### 2.4. Analysis of Cytokine Profiles in Plasma

The Bio-Plex multiplex cytokine assay (Bio-Rad Inc., Hercules, CA, USA) was used to measure 27 cytokines and chemokines in the blood samples with flow cytometric analysis (Appendix A).

### 2.5. Statistical Analysis

All analyses were conducted using the SPSS statistical software (version 26, IBM Corp., Armonk, NY, USA). Continuous data were presented as the median and interquartile range (IQR). Two-sided *p* < 0.05 was considered statistically significant. The Mann–Whitney U test was used to analyze the continuous and categorical variables to compare patients and controls.

## 3. Results

### 3.1. Patient Demographics

There were 13 patients (12 boys) in the WS group, with a median age of 5 months (IQR, 4.8–6.0). There were 11 symptomatic groups, with three cases of tuberous sclerosis complex (TSC), trisomy 21, Leigh syndrome, and periventricular leukomalacia (Table 1). ACTH therapy and vigabatrin were administered in seven and six patients, respectively. In three patients, a combination of ACTH and vigabatrin was administered.

Six patients displayed good prognosis, normal development, and mild developmental delay. The remaining six patients had severe developmental delay. Moreover, six patients with severe prognosis could not hold their neck up.

As control cases, we enrolled cryptogenic groups comprising 10 participants (five boys), with a median age of 6 months (IQR, 4.8–36). There were no statistically significant differences in age and sex between the patients and the control group (*p* = 0.07).

### 3.2. Intracellular Cytokine Expression

The percentage of IL-1β- and IL-1RA-positive monocytes in the WS group were significantly higher than that in the control group. Representative graphs and histograms of flow cytometry of monocytes (Figure 1a,b) clearly showed the difference in the percentage of IL-1β between the control and WS groups.

In addition, the intracellular interferon (IFN)-γ of CD8+ T cells in the WS group was found to be at a significantly higher level than that in the healthy control group. Table 2 summarizes the expression data obtained.

The WS group without sequelae had significantly higher levels of intracellular IFN-γ and interleukin (IL-) 6 in the CD8+ T and CD4+ T cells than the group with sequelae. The number of IL-1β- and IL-1RA-positive monocytes in the group with sequelae was higher than that in the group without sequelae, but these differences were not significant.

The total percentage of monocytes (CD14+), CD4+ T cells (CD3+, CD4+, CD25+, CD69+), CD8+ T cells (CD3+, CD8+, CD25+, CD69+), B cells (CD3+, CD19+), NK cells (CD3−, CD56+), and NKT- cells (CD3+, CD56+) were detected by immunofluorescence phenotyping, and we found no difference between patients with WS and controls.

No significant differences were found between the patients in the cryptogenic and those in the symptomatic groups.

### 3.3. Plasma Cytokine Levels

We compared the plasma levels of 27 cytokines and chemokines of the two groups. IL-6 levels were significantly higher in the control group (Appendix A).

## 4. Discussion

### 4.1. Pincipal Findings

We observed significantly higher levels of IL-1β- and IL-1RA-positive monocytes and IFN-γ-positive CD8 T-cells in the WS group compared with the control group.

To the best of our knowledge, this was the first study to examine the cells associated with the origin of cytokines using flow cytometry analysis in patients with WS; flow cytometry analysis in patients with intractable epilepsy has been performed [10]. The intracellular analysis of patients with Dravet syndrome revealed increased levels of pro-inflammatory cytokines, including IL-1β, in cells following the stimulation of monocytes with an in vitro vaccine [26]. In addition, the level of intracellular IL-1β in peripheral monocytes, but not the serum, of pediatric drug-resistant epilepsy was higher than that of controls. Furthermore, the level of intracellular IL-1β was correlated with the frequency of seizures, and we have confirmed the association between the clinical features and intracellular IL-1β levels [10].

Despite the flow cytometry results, there were no significant differences in the plasma IL-1β levels, consistent with previous reports [13,14,16]. As mentioned above, IL-1β in the blood was highly unstable, and IL-1β levels were frequently seen in the normal range, even with highly active systemic autoinflammatory diseases [28]. Moreover, IL-1β detection using standard techniques was difficult [29] owing to its presence in microvesicles [23]. In vitro studies by flow cytometry may allow the detection of IL-1β in intractable epilepsy.

This study and a previous study on DREs [10] consistently reported significantly higher levels of IL-1RA. Moreover, Shiihara et al. found significantly higher levels of serum IL-RA [13]; however, similar findings could not be detected in this study and in the previous study [14]. Researchers have reported a discrepancy between the intracellular cytokine levels in monocytes and in the plasma. Despite limited comparative analyses of intracellular cytokines and plasma levels, a similar trend has been observed in adult patients with intractable mesial temporal lobe epilepsy [25]. The underlying reasons for this discrepancy have not been completely elucidated as cytokines are expressed in various peripheral and central neurons.

In this study, patients with WS had increased levels of IFN-γ in their CD8+ T-cells than the healthy controls. Interestingly, WS patients with sequelae had significantly lower levels of intracellular IFN-γ and IL-6 in their CD8+ T and CD4+ T cells, respectively, than those without sequelae. However, the levels of IL-1β and IL-1RA were not significantly different. Both IFN-gamma and IL-6 are considered pro-inflammatory cytokines; however, they have dual inflammatory [30] and anti-inflammatory properties. The intracellular cytokine analysis of CD4+ T cells in adult patients with mesial temporal lobe epilepsy revealed elevated IL-6 [31] and IFN-γ levels [25]. IFN-γ exerts neurotoxic effects in specific epileptic syndromes, such as TLE [31]. However, whether these cytokines act in a neuroprotective or neurotoxic manner in the pathogenesis of epilepsy remains unknown.

In the WS group, serum IL-6 levels were in the normal range, similar to that in a previous report [5,14]. However, in the control group of this study, IL-6 levels were high, and a significant difference was confirmed. It is possible that the control group included cases with high levels of IL-6, such as those with an allergic constitution.

TNF-α has been shown to be associated with epilepsy [3], with Liu demonstrating elevated serum TNF-α levels. In WS, Liu et al. reported that symptomatic WS patients had higher serum TNF-α levels than cryptogenic WS patients. However, no significant difference was observed in this study. This might be due to the small sample size or due to the fact that accurate levels may not have been captured due to the short half-life [32].

Conventionally, resistance to treatment varies depending on the pathogenesis of WS, and recent reports have shown that patients with WS caused by structural acquired diseases (such as neonatal stroke and hypoxic-ischemic encephalopathy) are more responsive to ACTH treatment than patients with congenital disease [33]. In the present study, there was a trend toward a better prognosis in the group with no identifiable etiologies and in cases of tuberous sclerosis treated with vigabatrin. However, cytokines were not significantly different between the two groups, and past studies of cytokines have not yielded a unifying theory. Therefore, it is possible that immunogenicity and inflammatory pathways might also be related to the etiology of WS, but this is not evident at present.

Recent studies on human and mouse models have implicated the mechanisms that mediate the infiltration of cells from the periphery to the central nervous system (CNS) in the pathogenesis of epilepsy [34]. One study reviewed the relevance of the peripheral nervous system and CNS in the etiology of epilepsy [35]. We speculated that cell infiltration was not only a secondary destructive event associated with blood–brain barrier failure, but was also a non-disruptive event promoted by various mediators produced by the neurovascular unit [30]. Previous studies have shown that the knockout of CCR2, the receptor for CCL2, nearly abolished the kainic acid (KA)-induced elevation of IL-1β [36] and that activated resident microglia and infiltrating macrophages were the source of IL-1 elevation after KA-induced seizures, as evidenced by IL-1 expression in the microglia and monocytes. Our current study on pediatric intractable epilepsy, including WS, detected higher levels of intracellular IL-1β in peripheral monocytes, but not in other cells. In the pathogenesis of WS, the peripheral immunological reaction may influence the CNS, and peripheral monocytes producing IL-1β and IL-1RA are associated with CNS damage in WS.

### 4.2. Limitations

Owing to our small sample size, we could not perform statistical analysis for each etiology or drug, such as vitamin B6, ACTH, and vigabatrin. Moreover, vitamin B6 and anti-epileptic drugs could influence the immune parameters [37,38]. Vitamin B6 (or AEDs) may have an effect on cytokine regulation in West syndrome. In this study, we were not able to compare patients with and without vitamin B6 (or AEDs).

Although we examined the immune cells in the periphery, we did not analyze those in the spinal fluid or central tissues. Lastly, it was impossible to determine whether cytokines migrated during epileptic seizures or whether an inflammatory response was causing the seizures. Thus, the present study was exploratory, and the present results are suggestive but not definitive. Further studies are required to examine the mechanism by which peripheral immune responses affect the central immune system in WS pathogenesis.

## 5. Conclusions

Peripheral immune cells, such as monocytes, that produce IL-1β might be responsible for the pathogenesis of WS. Immunological methods using flow cytometry could facilitate the measurement of IL-1β levels, which remains challenging to assess, thereby contributing to immunological studies of epilepsy.

## Figures and Tables

**Figure 1 jcm-11-00447-f001:**
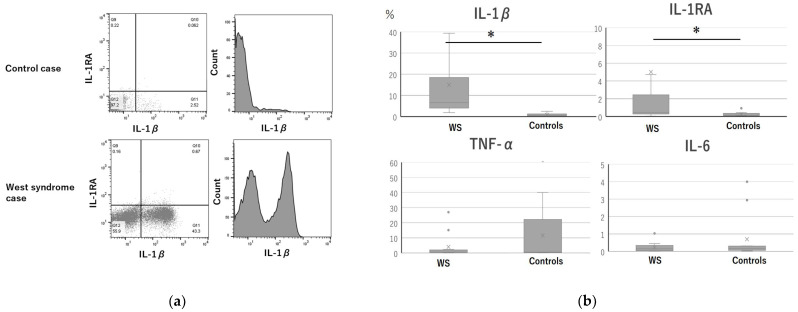
(**a**) An example of a plot of flow cytometry results for IL-1β and IL-1RA and the histogram of CD14-positive monocytes, analyzed by flow cytometry in patients with West syndrome and the control group. An increase in IL-1β production could be observed in patients with West syndrome. IL; interleukin, WS; west syndrome, TNF; Tumor necrosis factor (**b**) Intracellular cytokine expression in monocytes. Blood samples were obtained from 13 patients with West syndrome and 10 healthy controls. The boxplot extends from the 25th to the 75th percentile. The whiskers represent the maximum value on the top and the minimum value on the bottom. “×” represents the mean, and the line in the middle box represents the median. * *p* < 0.05.

**Table 1 jcm-11-00447-t001:** Clinical features and diagnosis of pediatric patients with West syndrome (*n* = 13).

Case	Sex	Age (Month)	Cause of West Syndrome	Antiepileptic Drugs Taken During Blood Sampling	Antiepileptic Drugs Administered After Blood Sampling	Outcomes
1	M	5	None	Vit.B6	ACTH, VGB	ND
2	F	6	None	Vit.B6	ACTH, VGB	ND
3	M	6	Tuberous sclerosis	None	VGB	ND
4	F	4	Tuberous sclerosis	ZNS	VGB	MD
5	M	5	Tuberous sclerosis	ZNS	VGB, VPA	MD
6	M	5	Cerebral infarction	ZNS	ACTH	MD
7	M	3	Lissencephaly	VPA, ZNS	VGB, Keto milk	SD
8	M	5	Focal cortical dysplasia	Vit.B6, ZNS, LEV	ACTH	SD
9	M	3	Unexplained brain atrophy	Vit.B6	ACTH, VGB, LTG	SD
10	M	5	Periventricular leukomalacia	VPA, ZNS	ACTH, Clonazepam	SD
11	M	7	Trisomy 21	TPM, VPA	VGB	SD
12	M	7	Leigh syndrome	ZNS, LEV	None	SD
13	M	5	Cerebral infarction	Vit.B6	ACTH	MD

M: male, F: female, ACTH: adrenocorticotropic hormone, ZNS: zonisamide, VPA: valproic acid, LEV: levetiracetam, VGB: vigabatrin, Vit.B6: vitamin B6, TPM: topiramate, ND: normal development, MD: mild developmental delay, SD: severe developmental delay.

**Table 2 jcm-11-00447-t002:** Comparison of intracellular cytokine levels between patients with West syndrome and the control participants.

	WS Group		Control Group		*p* Value
Monocytes					
IL-1β	6.5	(4.0, 18.4)	0.75	(0.4, 1.2)	0.000 *
IL-1RA	0.47	(0.28, 2.4)	0.35	(0.14, 0.43)	0.039 *
IL-6	0.14	(0.04, 0.34)	0.15	(0.02, 0.30)	0.48
TNF-α	2.0	(0.55, 2.3)	0.55	(0.35, 22.2)	0.51
CD4+ T cells					
IFN-γ	0.01	(0, 0.86)	0.02	(0.01, 0.64)	0.62
Granzyme A	0.42	(0.18, 0.73)	0.58	(0.33, 0.84)	0.76
IL-17	0.04	(0.02, 0.5)	0.09	(0.12, 0.46)	1.00
IL-10	0.04	(0.03, 1.29)	0.19	(0.11, 0.30)	0.59
IL-1β	0.32	(0.14, 0.84)	0.51	(0.19, 1.99)	0.49
IL-1RA	2.4	(0.80, 6.2)	3.7	(0.7, 8.4)	0.29
IL-6	0.05	(0.01, 0.21)	0.05	(0.01, 0.07)	0.38
TNF-α	0.3	(0.1, 0.42)	0.25	(0.21, 0.48)	0.96
CD8+ T cells					
IFN-γ	4.2	(0.38, 6.2)	1.9	(0.12, 3.7)	0.02 *
Granzyme A	2.3	(0.58, 9.5)	1.9	(0.60, 5.1)	0.26
IL-17	0.01	(0, 0.08)	0	(0, 0.04)	0.62
IL-10	0.02	(0.01, 0.26)	0.1	(0, 0.16)	0.89
IL-1β	1.8	(0.37, 1.9)	0.26	(0.09, 0.26)	0.29
IL-1RA	1.5	(1.1, 22.9)	4.1	(4.9, 0.3)	0.68
IL-6	0.24	(0, 1.3)	0.24	(0, 1.3)	0.34
TNF-α	0.3	(0, 0.16)	0.55	(0.02, 0.23)	0.71
NKT-like cells					
IFN-γ	0.04	(0.02, 0.07)	0.12	(0, 0.13)	0.690
Granzyme A	0.0	(0, 0.06)	0.01	(0, 0.02)	0.847
NK cells					
IFN-γ	0.69	(0.58, 1,3)	0.95	(0.21, 4.2)	0.405
Granzyme A	0.02	(0.01, 0.09)	0.28	(0.05, 0.4)	0.131
B cells					
IFN-γ	0.26	(0.01, 0.76)	0.25	(0.02, 0.47)	0.372
Granzyme A	0.02	(0.01, 0.04)	0.25	(0.02, 0.47)	0.846

WS; West syndrome, IFN; interferon, IL; interleukin, TNF; tumor necrosis factor, Data are presented as medians (interquartile range). * *p* < 0.05.

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
