# Peer review of "Involvement of Peripheral Monocytes with IL-1β in the Pathogenesis of West Syndrome"

_jcm, 2022, doi:10.3390/jcm11020447_

Round 1
Reviewer 1 Report
The objective of the manuscript was to examine the intracellular cytokine profiles of peripheral blood cells collected from 13 patients with WS using flow cytometry and measured their serum cytokine levels. The authors found a higher percentages of interleukin-1β (IL-1β), interleukin-1 receptor antagonist (IL- 20 1RA)-positive monocytes, and interferon gamma (IFN-γ) in CD8+ T cells. There was no correlation 23 between the ratios of positive cells and the serum levels of a particular cytokine in WS patients.
The justification to study cytokines in WS is superficial. The introduction only focus on IL-beta. More elaboration of the role of cytokine in WS is needed if the goal is focusing in cytokine profiles. What happened with other cytokines in WS, why focus on IL-1beta? How the IL-beta is affected by the course of the disease or treatment? How and when the IL-beta show high levels compare with controls?
Also, CSF samples should be included in the analysis to determine peripheral and neural IL-1 related with WS.
How antiepileptic drugs of VIT B6 were analyzed as confounding factor?
Any difference among different causes of WS? A table showing values in each patient has an important value to further studies (e.i. meta-analysis etc...)
Other cytokines should be described in the main text and not as supplementary information.
Is there any correlation between cell-surface makers CD3, CD4, and CD8, CD14, CD25, and 102 CD69 and cytokine levels?
What are the levels of those cytokine in those patients during interictal periods?
What is the discussion about IP-10 and IL-6 levels? This is a contradiction with some statement in discussion (line 170)
Explanation of TNF difference among groups is needed.
Author Response
Please see the attachment." in the box if you only upload an attachment

Reviewer 2 Report
Major
- The authors should provide literature review of older studies where they studies cytokines profile of patients with West Syndrome before and after treatment with ACTH.
- The authors should discuss and compare findings of this studies with other studies where they explored the role of immunogenicity and inflammatory pathways for pathogenicity of West Syndrome.
Minor points
- The authors should consider changing following line to make it more specific than generalization. "some cytokines have been associated with epilepsy".
Author Response
Please see the attachment

This manuscript is a resubmission of an earlier submission. The following is a list of the peer review reports and author responses from that submission.